# α-Zirconium(IV) Phosphate: Static Study of ^225^Ac Sorption in an Acidic Environment and Its Kinetic Sorption Study Using ^nat^Eu as a Model System for ^225^Ac

**DOI:** 10.3390/ma16175732

**Published:** 2023-08-22

**Authors:** Lukáš Ondrák, Kateřina Ondrák Fialová, Martin Vlk, Karel Štamberg, Frank Bruchertseifer, Alfred Morgenstern, Ján Kozempel

**Affiliations:** 1Department of Nuclear Chemistry, Faculty of Nuclear Sciences and Physical Engineering, Czech Technical University in Prague, 115 19 Prague, Czech Republic; 2Joint Research Centre, European Commission, Karlsruhe, Germany

**Keywords:** zirconium, phosphate, actinium-225, bismuth-213, sorption, kinetics

## Abstract

Zirconium phosphate (ZrP), especially its alpha allotropic modification, appears to be a very promising sorbent material for the sorption and separation of various radionuclides due to its properties such as an extremely high ion exchange capacity and good radiation stability. Actinium-225 and its daughter nuclide ^213^Bi are alpha emitting radioisotopes of high interest for application in targeted alpha therapy of cancer. Thus, the main aim of this paper is to study the sorption of ^225^Ac on the α-ZrP surface and its kinetics, while the kinetics of the sorption is studied using ^nat^Eu as a non-radioactive homologue of ^225^Ac. The sorption properties of α-ZrP were tested in an acidic environment (hydrochloric and nitric acid) using batch sorption experiments and characterized using equilibrium weight distribution coefficients *D*_w_ (mL/g). The modeling of the experimental data shows that the kinetics of ^225^Ac sorption on the surface of α-ZrP can be described using a film diffusion model (FD). The equilibrium weight distribution coefficient *D*_w_ for ^225^Ac in both hydrochloric and nitric acid reached the highest values in the concentration range 5.0–7.5 mM (14,303 ± 153 and 65,272 ± 612 mL/g, respectively). Considering the results obtained in radioactive static sorption experiments with ^225^Ac and in non-radioactive kinetic experiments with ^nat^Eu, α-ZrP seems to be a very promising material for further construction of a ^225^Ac/^213^Bi generator.

## 1. Introduction

Zirconium phosphates have been extensively studied since the 1950s [1,2]. In recent decades, they have been reported to have many applications, including as catalysts, drug delivery agents, anticorrosive agents, flame retardants, and ion exchangers [3,4,5,6,7,8]. The extensive interest in these materials across various scientific fields can be attributed to their excellent physicochemical properties, including exceptional ion exchange capability, thermal and radiation stability, biocompatibility, numerous active sites on the surface, and easy functionalization that leads to new structures with various applications [9,10]. Although earlier studies mainly focused on the use of zirconium phosphates as catalysts, recent interest has been centered on their exceptional ion exchange properties [11,12,13].

The investigation of the properties and structure of the α-ZrP molecule was aided by the Clearfield research group, who first synthesized zirconium(IV) phosphate in its alpha allotropic modification (α-ZrP) [1,14,15,16,17,18]. The molecule is comprised of zirconium atoms that are slightly positioned above or below its midplane. Cross-linking occurs from three oxygen atoms that are attached to another zirconium atom, from each phosphate group. The interlayer distance of α-ZrP is 0.76 nm. The thickness of the α-ZrP layer is 0.66 nm, and the remaining 0.1 nm is attributed to water molecules present in the interspace between individual α-ZrP layers [15]. Covalent bonds are formed between the atoms within the individual layers, while the interaction between neighboring layers is based on van der Waals forces [17].

In the case of using of zirconium phosphates as an ion exchanger, α-ZrP is the best known and the most promising due to its high selectivity for trivalent cations [18]. Numerous studies on ion exchange separation of radioactive metals were conducted across the periodic table of elements, including from the light elements to the super-heavy elements, such as actinides [19,20,21]. The outstanding suitability of α-ZrP as an ion exchanger makes it applicable in the separation of medically significant radionuclides [13] or in the field of radioactive waste reprocessing [11]. Previous studies have explored the sorption of various metal ions using zirconium phosphate in crystalline or amorphous form, or in a composite sorbent based on a polyacrylonitrile matrix. The crystalline form of zirconium phosphate is specifically suitable for separating trivalent lanthanoids and actinoids during nuclear fuel reprocessing. The purpose of this material, especially nanocrystalline α-ZrP, was demonstrated in the separation of Eu(III)/Am(III) in nuclear fuel in the pH range of 0–3 in nitric acid. The highest separation factor of 400 was achieved at pH 1 [11]. Zirconium phosphate in its amorphous phase can also be utilized to separate different metals, particularly bivalent (Cu(II), Mn(II), Ni(II), Zn(II) and Pb(II)) and tetravalent (Th(IV)). The ability of this sorbent to separate stated metal ions was tested in various media, including 0.2 M and 0.02 M NH_4_NO_3_, HNO_3_, HClO_4_, and CH_3_COOH. The distribution coefficients of bivalent metal ions in the eluents depend on the pH and ionic strength. When the pH drops below the eluent’s pK value, the acid groups become mostly nonionic, leading to a decrease in the apparent capacity and distribution coefficients. The distribution coefficient shows that amorphous zirconium phosphate demonstrates higher selectivity towards Pb(II), Ni(II), and Mn(II). In 0.02 M acetic acid media, Pb(II) and Cu(II) show higher distribution coefficients than in distilled water. Except for Ni(II), Mn(II), and Th(IV), all the evaluated metal ions exhibit higher distribution coefficients in electrolyte media compared with distilled water [22]. Additionally, the amorphous form of zirconium phosphate served as a composite sorbent, based on a polyacrylonitrile matrix, to separate Co(II), Nd(III), and Dy(III). By using a single column with 1 mM nitric acid, the purity of simulated leachate increased to 87.9% for Co, 96.4% for Nd, and 40% for Dy at different stages of effluent [23].

This study aims to investigate the kinetic and equilibrium sorption of ^225^Ac on α-ZrP particles in the acidic environment of nitric and hydrochloric acid with future potential application in the separation of ^213^Bi as a medically relevant therapeutic radionuclide in targeted alpha therapy (*TAT*), which, together with other medical branches such as supramolecular nanochemotherapy, belongs to the modern therapeutic options for the treatment of oncological diseases [24,25]. Considering that ^213^Bi has been used in clinical trials, primarily on patients with leukaemia, bladder cancer, neuroendocrine tumors, melanoma, glioma, and lymphoma. Due to the very positive results of the clinical studies mentioned above, it can be assumed that this radionuclide will soon become established in normal clinical practice and the demand for its availability will increase rapidly [26,27,28,29,30,31]. The equilibrium sorption properties of α-ZrP were tested in radioactive experiments using ^225^Ac, while the kinetics of the sorption process on the α-ZrP particles was tested using ^nat^Eu as a non-radioactive homologue of ^225^Ac.

## 2. Materials and Methods

### 2.1. α-ZrP Preparation

α-ZrP was prepared by mixing 6.4 g of zirconium(IV) oxychloride octahydrate (Sigma-Aldrich, Darmstadt, Germany) in 20 mL of demineralized water solution and 55 g of sodium dihydrogen phosphate monohydrate (Sigma-Aldrich, Germany) in 40 mL of 3 M ultrapure hydrochloric acid solution (Sigma-Aldrich, Germany). The sodium dihydrogen phosphate monohydrate solution was added dropwise to a solution of zirconium(IV) oxychloride octahydrate at 80 °C and the mixture was stirred throughout (heating and stirring of the mixture was performed using an IKA C MAG HS 7 (IKA-Werke GmbH & Co. KG, Staufen im Breisgau, Germany)). The mixture was then refluxed at 80 °C for 30 h. Finally, the reaction mixture was left at room temperature for 2 days without stirring to promote precipitation [10]. Finally, the precipitate was filtered and washed with 200 mL of 3 M phosphoric acid (Sigma-Aldrich, Germany), and deionized water to pH 3 was reached. The precipitate was dried to a constant weight loss at 60 °C using a Binder 9630-0002 VD 56 Standard Vacuum Drying Chamber (Binder, Tuttlingen, Germany).

The prepared α-ZrP was characterized using various analytical methods such as infrared spectroscopy, X-ray powder diffraction, thermogravimetry, differential thermal analysis, and scanning and transmission electron microscopy in the work [32].

### 2.2. Kinetic Study of ^nat^Eu Sorption on the Surface of α-ZrP

The kinetics of sorption on α-ZrP particles was studied in a non-radioactive experimental mode using ^nat^Eu as a suitable and available ^225^Ac homologue due to the lack of a stable isotope of actinium. The most typical analog of Ac is considered to be La, although the use of Eu as a suitable analog of Ac due to its physicochemical properties corresponding to those of a typical lanthanide can be found in the literature in the context of sorption studies of actinium or actinides in general [33,34,35,36].

The kinetic study was performed with 1 g of prepared α-ZrP. The total volume of the solution of 0.1 M ultrapure hydrochloric acid (Sigma-Aldrich, Germany) with ^nat^Eu (in the form of europium(III) nitrate pentahydrate (Sigma-Aldrich, Germany)) at a concentration of 10,000 ppb was 100 mL (V/m ratio 100 mL/g). The time of addition of the sorbent to the Eu solution was marked as time t = 0. The sorbent suspension was continuously stirred throughout the experiment using an IKA NANOSTAR 7.5 digital (IKA-Werke GmbH & Co. KG, Germany). To determine the instantaneous residual concentration of Eu in the suspension, 0.2 mL aliquots of the stirred suspension were taken at times 0, 0.5, 1, 5, 10, 15, 20, 25, 30, 45, 60, 90, and 120 min. All aliquots taken were filtered through a microfiber glass filter and then diluted 30× with 5% solution of ultrapure nitric acid (Sigma-Aldrich, Germany) for ICP-MS analysis of Eu content.

ICP-MS analysis of the prepared samples was performed on an Agilent 7500 Series ICP-MS (Agilent Technologies Inc., Santa Clara, CA, USA) with a radiofrequency source power of 1550 W, a sampling rate of 0.1 rps, a carrier gas flow rate of 1.0 L/min, and a sampling cone distance of 7 mm in collisionless gas mode. The data were processed using the MassHunter program (version B.01.01).

Finally, the experimental data were evaluated using several models for two-phase systems. The kinetic models used are summarized in Table 1 [37]. The models reflect the following different rate-controlling processes: mass transfer (DM), film diffusion (FD), diffusion in inert layer (ID), diffusion in reacted layer (RLD), chemical reaction (CR), and gel diffusion (GD).

The following balance and equilibrium equations apply:(13)$$\frac{dq}{dt}=-r\cdot \frac{dc}{dt}$$
and
(14)$$q=r\cdot \left(c-c_{0}\right)+q_{0}$$
where *c*_0_ is the starting concentration of the component in the aqueous phase. And finally:(15)$$q^{*}=K_{d}\cdot c$$
and
(16)$$c^{*}=\frac{q}{K_{d}}\;,$$
where *K_d_* is the distribution coefficient.

The procedure of experimental data evaluation by means of the above-mentioned models has been described and demonstrated in detail by Suchánková et al. [38]. For this evaluation, in the course of which the values of the total mass transfer coefficients, e.g., *K_FD_*, are sought, the Newton–Raphson multidimensional non-linear regression method, combined with the solution of the differential equation under given boundary conditions (by means of the Runge–Kutha method), was used. The code P60.fm (in the software product FAMULUS (version 3.5), code package STAMB 2016) was used. Of course, the quantities *K_d_* (i.e., distribution coefficient), the mean radius of solid phase particle, *R*, and the initial concentrations, *c*_0_ and *q*_0_, must be known. The input data for the calculations of the kinetic model parameters and the modeling of the ^225^Ac sorption kinetics in the case of this paper were as follows: *K*_D_ = 2.51 × 10^3^ L/kg; *C*_0_ = 6.80 × 10^−5^ M; *q*_0_ = 0; *C*_exp_ = f(*t*); *r* = *V*/*m* = 100 mL/g.

The quantity WSOS/DF (weighted sum of squares of differences divided by number of degrees of freedom) is used as the criterion of goodness-of-fit (for this particular criterion, the fit is considered acceptable if 0.1 < *WSOS/DF* < 20). Its calculation is based on the *χ*^2^ test calculated according to Equation (17):(17)$${Χ}^{2}=\sum \frac{{\left(SS_{x}\right)}_{i}}{{\left(s_{q}\right)}_{i}^{2}},\;i=1,\;2,\;3,\ldots ,n_{p},$$
where (*SS_x_*)*_i_* is the *i*-th square of the deviation of *i*-th experimental value from the corresponding calculated one, and (*s_q_*)*_i_* is the estimate of standard deviation (uncertainty) of the *i*-th experimental point.

Then the *WSOS/DF* is obtained by means of Equation (18):(18)$$WSOS/DF=\frac{{Χ}^{2}}{n_{d}},$$
where *n_d_* is defined by means of Equation (19):(19)$$n_{d}=n_{p}-n,\;$$
where *n_d_* is the number of degrees of freedom, *n_p_* is the number of experimental points, and *n* is the number of model parameters sought during the regression procedure.

The experimental kinetic dependence was fitted step by step by all six models (shown in Table 1), resulting in the values of the goodness-of-fit criterion, *WSOS/DF*, and of the values of the overall kinetic coefficients.

### 2.3. Batch Experiments of ^225^Ac Sorption on the Surface of α-ZrP

The samples were mixed using an orbital shaker, KS250 basic (IKA-Werke GmbH & Co. KG, Germany), during the experiment investigating the sorption of ^225^Ac and ^213^Bi under static conditions in an acidic environment.

A phase separation of α-ZrP in nitric acid suspension to prepare a sample for gamma counting was performed using a coil of a Whatman GF/C microfiber glass filter (GE HealthCare, Chicago, IL, USA) in a 5 mL polypropylene tip. The activity of the samples was measured using an Ortec HPGe–Dspec Junior 2.0 (Ortec, Oak Ridge, TN, USA) gamma spectrometer. The measured spectra were evaluated using MAESTRO Multichannel Analyzer Emulation Software (version 7.01) (Ortec, United States). The evaluation of the measured gamma spectra was performed using a line of ^213^Bi with an energy 440 keV. Gamma spectrometry of all experimentally obtained samples was performed after establishing a permanent radioactive equilibrium between ^225^Ac and ^213^Bi.

Actinium-225 was purchased dry from the Joint Research Center, European Commission in Karlsruhe, Germany. Before use, ^225^Ac was dissolved in 0.1 M hydrochloric acid.

Batch experiments were performed by mixing a defined amount of prepared α-ZrP as an ion exchanger (100 mg) with a defined amount of nitric acid solution in the concentration range of 0.0001 to 1 M (10 mL). A defined amount of ^225^Ac (10 μL of stock solution in 0.1 M hydrochloric acid, corresponding to approximately 10 kBq of ^225^Ac, a molar concentration of Ac^3+^ of approximately 2 pmol/L) was added to the prepared α-ZrP in nitric or hydrochloric acid suspensions. The prepared suspensions were shaken for 2 h and then filtered through a glass fiber filter. For each sample, a corresponding standard solution was prepared (the only difference between the sample and the standard solution was the presence/absence of the α-ZrP in the vessel). Thanks to this experimental design, the sorption of 225Ac on the vessel walls had no influence on the values of the experimentally obtained weight distribution coefficients *D_w_*. Aliquots of 1 mL volume were prepared from all samples and standards for gamma counting with the above-mentioned gamma spectrometer.

The weight distribution coefficients for ^225^Ac in each used concentration of nitric or hydrochloric acid were calculated by Equation (20):(20)$$D_{w}=\frac{A_{st}-A}{A}\cdot \frac{V}{m}\;,\;$$
where *A_st_* is activity of standard [imp/s], *A* is activity of filtrate [imp/s], *V* is volume of aqueous phase [ml] and *m* is weight of α-ZrP [g].

## 3. Results and Discussion

### 3.1. Kinetic Study of ^nat^Eu Sorption on the Surface of α-ZrP

Experimental data of ^nat^Eu kinetic sorption on the α-ZrP surface are given in Table 2 as a fraction *F* of Eu cations remaining in the solution in time.

The fraction *F* was defined as:


(21)
$$F=\frac{c}{c_{0}}.$$


As can be seen from the kinetics data presented in Table 2, the kinetics of Eu sorption on the surface of α-ZrP is quite fast and in 5 min, 93% of the Eu cations are sorbed by α-ZrP. Based on the assumption that Eu is homologous to Ac, this makes α-ZrP a promising material in the case of ^225^Ac sorption, and modeling of the kinetic process will follow, with the aim of finding the control step of the sorption process. The results of the values of the goodness-of-fit criterion, WSOS/DF, and of the values of the overall kinetic mass transfer coefficients *K*_model_ resulting from the stepwise fitting of the experimental kinetic data using all six models mentioned and defined above, are presented in Table 3. The two most promising models (FD and RLD) are shown graphically in Figure 1a,b.

According to the WSOS/DF values, it is obvious that there is only one model suitable for modeling and describing the experimental data, namely the FD model. It is the model derived under the assumption that diffusion in the liquid film covering the surface of the solid sorbent particle is the rate-controlling process. The low value of the standard uncertainty (about 0.05%) of the KFD quantity also confirms the suitability of the FD model. Considering the relationships for the FD model in Table 1, it is clear that the rate of kinetics is inversely proportional to the surface film thickness and the particle size of the sorbent. It is directly proportional to the value of the partition coefficient *K_d_*, which determines the value of the parameter *c**, i.e., the magnitude of the driving force of the transport process.

The obtained result for the FD model agrees well with the sorption kinetics on classical ion exchangers, especially on cation exchangers, under conditions characterized by sorption from solutions with a very low concentration of the monitored component [37]. Finally, the studied sorbent is generally considered to be a cation exchanger.

### 3.2. Batch Experiment of ^225^Ac Sorption on the Surface of α-ZrP

In Table 4 (results obtained in hydrochloric acid) and Table 5 (results obtained in nitric acid), the experimentally obtained results of ^225^Ac sorption on the particles of the α-ZrP static study, carried out according to the experimental setup described above, are presented as coefficients *D_w_*. The graphical representation of the results obtained is shown in Figure 2a,b.

The dependence of the weight distribution coefficient *D_w_* on the molarity *c* of the acid used, shown in Figure 2a,b, shows a sharp maximum in the concentration range around 0.005 M for both hydrochloric acid and nitric acid. As can be seen in Figure 2a,b, in the range of lower concentrations (from 0.0001 to 0.001 M) and higher concentrations (from 0.1 M to 1 M) for both acids, there is no visible or very little sorption of ^225^Ac. The values of the equilibrium weight distribution coefficient *D_w_* for ^225^Ac in both hydrochloric and nitric acid reached values 14,303 ± 153 mL/g and 65,272 ± 612 mL/g respectively, in the concentration range of 5–7.5 mM. Therefore, the concentration range around 0.005 mM seems to be appropriate for the next applications (e.g., ^225^Ac/^213^Bi radionuclide generator construction). However, a comparison of the results obtained with the two acids used clearly shows that nitric acid is a more suitable medium for the sorption of ^225^Ac on the surface of α-ZrP.

The following possibilities are offered to explain this phenomenon:(a)In the range of hydrochloric and nitric acid concentrations higher than 0.01 M, i.e., at pH < 2, the cationic sorption capacity of α-ZrP decreases significantly, as shown not only by our results from the evaluation of the corresponding titration curve [32], but also by data in the literature devoted to the properties of α-ZrP, e.g., [39].(b)Increasing the acid concentration increases the ionic strength, which generally leads to a decrease in the equilibrium constants, eventually also of *D_w_*.(c)The difference between the hydrochloric and nitric acid environments is probably due to the generally higher complexation efficiency of the chloride ligand compared with the nitrate ligand, resulting in a reduction of the cationic form in solution and a decrease of its sorption capacity. Unfortunately, the values of the stability constants for Ac^3+^-Cl^−^ and Ac^3+^-NO_3_^−^ complexation could not be found in the available literature. However, there is a significant difference in the values of the dissociation constants of these acids; in the case of hydrochloric acid, the pKa has value −6, and in the case of nitric acid, the pKa has value −1.32. Thus, the concentration of the ligand in the dissociated state is an order of magnitude higher in the case of hydrochloric acid than in the case of nitric acid, which may contribute to a higher complexation efficiency of the cationic form of ^225^Ac^3+^ and consequently to a lower sorption capacity of α-ZrP in the hydrochloric acid environment compared with the corresponding values in the nitric acid environment [40].(d)In the range of concentrations below 0.001 M, i.e., at pH > 3 and especially at pH > 4, a gradual hydrolysis of the cationic forms and an increase in the concentration of hydroxo-complexes in solution and a decrease in the sorption of the cationic forms can be expected.

## 4. Conclusions

The modeling of the experimental data shows that the kinetics of the model Eu sorption on the particles of α-ZrP can be described in the most satisfactory way using a model of film diffusion (FD). Taking into account the kinetic equations for FD, it is clear that the rate of kinetics is inversely proportional to the thickness of the surface film and the particle size of the sorbent (the particle size of the used particles was determined to be 150 nm in the previous work [32]). The results of the static sorption experiments show that the equilibrium weight distribution coefficients *D_w_* for ^225^Ac in both hydrochloric and nitric acid reached the highest values of 14,303 ± 153 mL/g and 65,272 ± 612 mL/g, respectively, in the concentration range from 5.0 to 7.5 mM. From the comparison of these two results obtained for the equilibrium weight distribution coefficients *D_w_* of ^225^Ac in both used acids, it can be concluded that nitric acid is a more promising environment for ^225^Ac sorption on the α-ZrP particles, and thus the use of nitric acid and α-ZrP can be considered as a very interesting combination for further research of ^225^Ac/^213^Bi generator construction. The developed material seems to be very promising even in comparison with other separation systems often described as generator systems in the literature. Compared to the extraction agents used, their use for the purpose of separating ^213^Bi from the parent ^225^Ac is described by some authors in the literature as a two-step process, where in the second step of the separation, the eluent is concentrated or the extractant molecule is removed and ^213^Bi is converted into a salt form [41,42]. Generator separation systems for the separation of ^213^Bi often appear in the literature, but they work as reverse systems, i.e., ^213^Bi is filtered on the columns while the parent ^225^Ac flows through the column without retention to the column material and the ^213^Bi is subsequently washed out with another eluent [42,43]. Even in comparison to these systems, the developed α-ZrP should be more advantageous, since it allows a classical separation in a direct arrangement. Compared with the above-mentioned separation systems, our developed material shows parameters and properties based on which it can be promised that the future generator system based on developed material combined with 5 mM nitric acid as eluent will give better results in terms of ^213^Bi acquisition rate, ease of carrying, and, of course, price.

## Figures and Tables

**Figure 1 materials-16-05732-f001:**
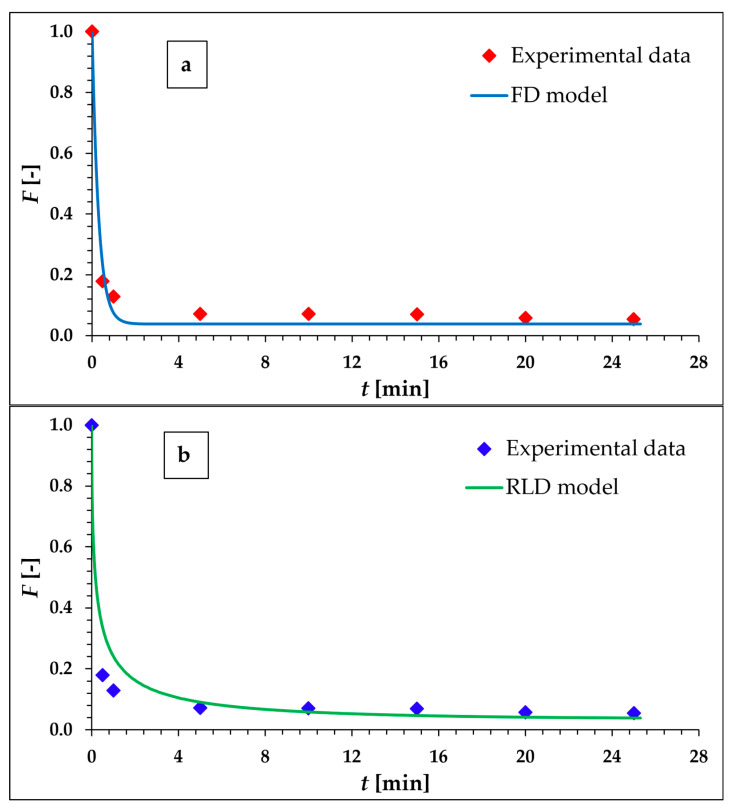
Evaluation of kinetic experimental data by means of FD (**a**) and RLD (**b**) kinetic models.

**Figure 2 materials-16-05732-f002:**
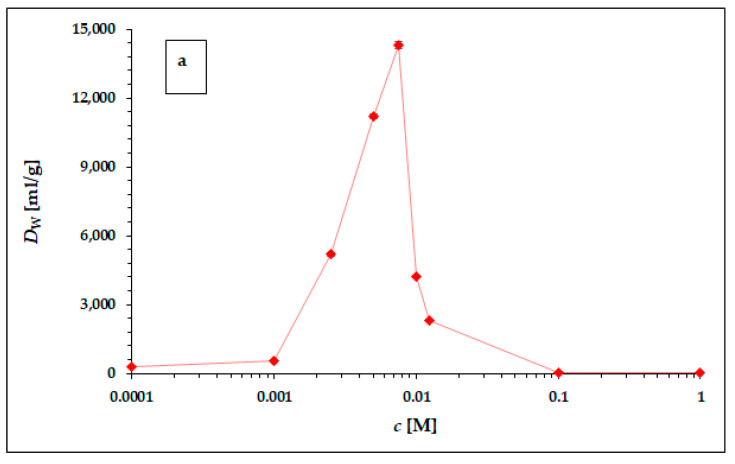
The dependence of ^225^Ac weight distribution coefficient *D_w_* in hydrochloric (**a**) and nitric (**b**) acid. The error bars corresponding to the deviations *σ*_D*w*_ presented in Table 4 and Table 5 are not visible due to the scale of the *y*-axis. The line chart was used for better orientation and following the dependence.

**Table 1 materials-16-05732-t001:** Kinetic models of sorption/extraction taking place in two-phase systems [37].

Control Process	Model	Differential Equation	
Mass transfer	DM	$\frac{dq}{dt}=K_{DM}\cdot \left(q^{*}-q\right)$	(1)
Film diffusion	FD	$\frac{dq}{dt}=K_{DM}\cdot \left(c-c^{*}\right)$	(2)
		$K_{ID}=\frac{3\cdot D}{\delta \cdot R\cdot h_{s}}$	(3)
Diffusion in an inert layer	ID	$\frac{dq}{dt}=K_{ID}\cdot \frac{\left(c-c^{*}\right)}{{\left\{1-\left(\frac{q}{q^{*}}\right)\right\}}^{-\frac{1}{3}}-1}$	(4)
		$K_{ID}=\frac{3\cdot D}{R^{2}\cdot h_{s}}$	(5)
Diffusion in a reacted layer	RLD	$\frac{dq}{dt}=K_{RLD}\cdot \frac{\left(q-q^{*}\right)}{{\left\{1-\left(\frac{q}{q^{*}}\right)\right\}}^{-\frac{1}{3}}-1}$	(6)
		$K_{RLD}=\frac{3\cdot D}{R^{2}\cdot h_{s}}$	(7)
Chemical reaction in reaction zone	CR	$\frac{dq}{dt=K_{CR}\cdot r_{CR}\cdot {\left\{1-\left(\frac{q}{q^{*}}\right)\right\}}^{\frac{2}{3}}}$	(8)
		$K_{CR}=\frac{3}{R\cdot h_{s}}$	(9)
		$r_{CR}=k_{CR}\cdot \left(c-c^{*}\right)$	(10)
Gel diffusion	GD	$\frac{dq}{dt}=K_{GD}\cdot \frac{{\left(q^{*}-q_{0}\right)}^{2}-{\left(q-q_{0}\right)}^{2}}{\left(q-q_{0}\right)}$	(11)
		$K_{GD}=\frac{D\cdot \pi^{2}}{2\cdot R^{2}}$	(12)

Key to the used symbols: *c*—concentration of the component in the aqueous phase at the time *t*; *c**—equilibrium concentration of the component in the aqueous phase corresponding to the concentration of the component in the sorbent at the time *t*; *q*—concentration of the component in the sorbent at the time *t*; *q**—equilibrium concentration of the component in the sorbent corresponding to the concentration of the component in the aqueous phase at the time *t*; *q*_0_—starting concentration of the component in the sorbent; *t*—time; *r*—volume ratio of aqueous to solid phase; *D*—diffusion coefficient of the component, *K_DM_*; *K_FD_*, *K_ID_*, *K_RLD_*, *K_CR_*, *K_GD_*—overall kinetic coefficients; *k_CR_*—kinetic coefficient of the chemical reaction; *r_CR_*—rate of the chemical reaction; *R*—mean radius of the solid phase particle; *h_s_*—specific mass of the solid sorbent; *δ*—thickness of the “liquid film” on the surface of the solid particle.

**Table 2 materials-16-05732-t002:** Values of fraction *F* of Eu cations remaining in the solution in time.

t [min]	*F* [-]
0	1.000
0.5	0.179
1	0.129
5	0.071
10	0.071
15	0.070
20	0.058
25	0.054

**Table 3 materials-16-05732-t003:** Values of goodness-of-fit criterion, WSOS/DF, and overall kinetic mass transfer coefficients, *K*_model_.

Model	WSOS/DF [-]	*K*_model_ [min^−1^]
DM	1.38 × 10^2^	(2.48 ± 0.01) × 10^−4^
FD	7.79 × 10^0^	(3.17 ± 0.01) × 10^2^
ID	4.14 × 10^2^	(2.21 ± 3.28) × 10^−1^
RLD	1.42 × 10^1^	(9.68 ± 0.03) × 10^−4^
CR	1.20 × 10^2^	(6.96 ± 20.7) × 10^2^
GD	2.42 × 10^3^	(2.56 ± 7.40) × 10^−5^

**Table 4 materials-16-05732-t004:** Values of experimentally obtained ^225^Ac weight distribution coefficients, *D_w_*, in hydrochloric acid and its deviation, *σ_Dw_*, calculated based on error propagation law.

c [M]	*D_w_* [mL/g]	*σ_Dw_* [mL/g]
0.0001	267	11
0.0010	537	15
0.0025	5181	56
0.0050	11,206	126
0.0075	14,303	153
0.0100	4240	51
0.0125	2293	56
0.1000	59	4
1.0000	45	4

**Table 5 materials-16-05732-t005:** Values of experimentally obtained ^225^Ac weight distribution coefficients, *D_w_*, in nitric acid and its deviation, *σ_Dw_*, calculated based on error propagation law.

c [M]	*D_w_* [mL/g]	*σ_Dw_* [mL/g]
0.0001	2662	28
0.0010	4482	42
0.0050	65,272	612
0.0100	19,486	187
0.1000	41	1
1.0000	11	1

## Data Availability

There are no further data provided.

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
