# Peer review of "α-Zirconium(IV) Phosphate: Static Study of 225Ac Sorption in an Acidic Environment and Its Kinetic Sorption Study Using natEu as a Model System for 225Ac"

_materials, 2023, doi:10.3390/ma16175732_

Round 1

Reviewer 1 Report (New Reviewer)

Ondrák et al. presenting an article title "α-zirconium(IV) phosphate: static study of 225Ac sorption in acidic environment and its kinetic sorption study using natEu as a model system for 225Ac".

the article looks good i have no comments. good luck further

Author Response

Reviewer 2 Report (New Reviewer)

This manuscript studied the adsorption and kinetics of 225Ac on the surface of α-ZrP, and natEu was used as the non-radioactive homologue of 225Ac. The model analysis of experimental data showed that the equilibrium weight partition coefficient Dw of 225Ac in both hydrochloric and nitric acid reached a maximum in the concentration range of 5.0 to 7.5 mM. This study contributes to the exploration of 225Ac/213Bi generator construction. However, there are still some problems in the manuscript. It is recommended that the manuscript be published in materials after completing revision.

1.      In the introduction, it is suggested to add the illustration of α-ZrP to make readers understand α-ZrP more clearly.

2.      In the introduction, the connection between the third and fourth paragraphs is too stiff, and it is suggested to modify and polish it.

3.      The α-ZrP preparation part of the article mentioned that the prepared α-ZrP was characterized, please add the characterization data.

4.      The formulas in the article are not arranged neatly, such as “Formulas 13 and 14”. Please check the manuscript for similar issues and make corrections.

5.      There are some problems in the pictures of the article, for example: “The pictures in Figure 1 were overlapped”. Please check the pictures carefully in the article and correct the deficiencies.

6.      The last sentence on page 10 is not complete. Please revise it.

7.      The format of references is not uniform. For example, the case of journals in Ref. 1 and Ref. 2 is not uniform. Please check the similar issues and revise.

8.      The introduction part of the article can add content related to the significance of the research, such as in cancer diagnosis and treatment. Here are many recent articles for reference: Chem. Soc. Rev. 2021, 50, 2839; Exploration 2021, 1, 21; Adv. Mater. 2023, DOI: 10.1002/adma.202304249.

Extensive editing of English language required

Author Response

Reviewer 3 Report (New Reviewer)

In general, the research conducted by the authors deserves attention in terms of relevance, scientific and practical significance and may be useful for researchers involved in the development of new materials for the isolation and separation of rare, noble, radioactive elements. Meanwhile, I have a number of questions and comments on the work: 1. The examples of extraction of 225Ac given in the experimental part are essentially ideal, but what do the authors think about the processes of extracting this element from mixed solutions? And what methods are supposed to carry out desorption to obtain pure 225Ac? In addition, in conclusion, it is necessary to clearly describe the prospects for the development of this area of research, as well as to compare the proposed method and reagent with the best samples that exist today. What are the advantages of the systems you have developed over the known ones? Price, availability of reagents, speed, ease of carrying out the extraction/desorption process, etc.

Author Response

This manuscript is a resubmission of an earlier submission. The following is a list of the peer review reports and author responses from that submission.

Round 1

Reviewer 1 Report

The paper entitled "α-zirconium(IV) phosphate: kinetic and static study of 225Ac sorption in acidic environment" describes the kinetic and equilibrium studies of Actinium-225 sorption on α-ZrP particles in acidic medium (nitric and hydrochloric acids).

The authors used a series of interesting mathematical models to investigate the kinetic studies of Actinium-225 sorption process of in acidic medium, but there are some shortcomings that should be clarified:

- the data presented in Table 1 are similar to those in Figure 1a

- the results of the experiments are very little explained

- the obtained results are not compared with those obtained for similar systems found in the literature. What are the advantages of ZrP particles compared to other systems?

Reviewer 2 Report

I would like to thank the authors for the effort that went into producing the research in such a good way, and for their success in selecting the research point that supports environmental conservation, public health and the economy.

There are some minor notes such as:

A table of abbreviations should be added before references

Ensure that all addresses are written uniformly

Thanks

Reviewer 3 Report

Congratilations on your good work. 

Reviewer 4 Report

The authors have assessed the ability of ZrP to be used as a solid phase material for the sorption of Ac-225 for future use as a Ac-225/Bi-213 generator solid support. ZrP was synthesized and kinetic studies were first performed with natural Eu as a surrogate. However no evidence was provided as to why Eu is a better analogue for Ac than La is. Please search the literature and provide references on when this approximation has also been used. The sorption modeled well after film diffusion.

I have attached the manuscript with highlights and comments for improvements.

I suggest using an English proofreading service to correct grammar and language.

I have attached the manuscript with highlights and comments for improvements.

Reviewer 5 Report

This review concerns manuscript titled: “α-zirconium(IV) phosphate: kinetic and static study of 225Ac sorption in acidic environment” submitted for consideration to the journal Materials. The manuscript discusses adsorption properties of zirconium phosphate sorbents toward 225Ac. After reading the manuscript, it is hard to recommend it for publication due to low quality and questionable significance. Especially, the authors wish to study adsorption properties of 225Ac but use europium as the adsorptive instead. No justification or prior precedent for such studies is provided. The authors claim Eu is a homologue of Ac but the latter belongs to actinides while the former is a lanthanide, and are in different groups too. Thus, the validity of kinetic studies with respect to describing kinetics of Ac sorption is doubtful. The present reviewer also dislikes the fact that the manuscript frequently refers to “kinetic studies of 225Ac” (see for example conclusion section), which seem deceiving since they are in reality kinetic studies of Eu but prospective readers will not know unless they read the experimental section in detail. The fitting of kinetic data, and the discussion following from it, also seems doubtful. First, the experiment is poorly designed since only two datapoints out of 8 are meaningful. 1 is always 1 by definition, and all fits start from 1, so it benefits the fit little. The points beyond 4 min are flat and benefit the fit little as well. The points in the range 0-4 min would inform fitting the most but there are only two, and they are rather similar, so effectively one point in that range. For these reasons the present reviewer doubts that (i) FD is in fact the best mechanism to descried the kinetics of Eu adsorption and (ii) the fitting procedure can meaningfully differentiate between the investigated models. As a side note, the symbol for zirconium phosphate is also rather particularly poorly chosen as ZrP since most prospective readers would expect it to related to zirconium phosphide.

krystalline

Founf

Round 2

Reviewer 1 Report

The paper can be published in present form.

Reviewer 4 Report

The authors have addressed my comments and suggestions sufficiently. Pity to hear about the lack of availability of La, but thank you for including some literature search on the use of Eu.